# Prevalence, incidence and associated factors of pneumonia among severely malnourished children hospitalized in Mulago National Referral Hospital, Uganda

**Harriet Nambuya** [1,2], **Ezekiel Mupere**[2], **Nicolette Nabukeera-Barungi**[2,3], **Paul Mubiri**[4], **Jolly General Kamugisha**[5‡], **Maren J. H. Rytter**[3], **Rebecca Nantanda**[6]

1 Department of Pediatrics and Child Health, Jinja Regional Referral Hospital, Jinja, Uganda, 2 Department of Pediatrics and Child Health, College of Health Sciences, Makerere University, Kampala, Uganda, 3 Department of Nutrition, Exercise and Sports, Faculty of Science, University of Copenhagen, Copenhagen, Denmark, 4 Department of Health Policy and Management, School of Public Health, Makerere University College of Health Sciences, Kampala, Uganda, 5 Department of Pediatrics and Child Health, Mwanamugimu Nutrition Unit, Mulago National Referral and Teaching Hospital, Kampala, Uganda, 6 Makerere University Lung Institute, College of Health Sciences, Makerere University, Kampala, Uganda

⊜ These authors contributed equally to this work.
‡ JGK also contributed equally to this work.
\* nambuyaharriet9@gmail.com

**Data availability statement:** Details data access are indicated below "The Ugandan act on Data

## Abstract

### Background

Pneumonia is still a burden in children hospitalized with severe acute malnutrition (SAM). However, there is limited published information on the magnitude and characteristics of those who present with or develop pneumonia during hospitalization. We determined the prevalence and incidence of pneumonia, and associated factors among children aged 6-59 months hospitalized with SAM.

### Methods

This study employed 2 study designs; cross sectional for prevalence of pneumonia and single arm retrospective cohort design for incidence of pneumonia. We used secondary data collected as part of the ProbiSAM trial that assessed the effect of Probiotics on diarrhea in children with severe acute malnutrition conducted at Mwanamugimu Nutrition Unit Mulago Hospital. Children aged 6-59 months hospitalized with SAM were assessed for prevalence and incidence of pneumonia. Prevalence of pneumonia was the proportion of pneumonia cases at the time of enrolment or within the first 48 hours of admission. 2)Incidence of pneumonia was the proportion who developed clinical signs of pneumonia for the first time during hospitalization among those who did not have pneumonia at enrolment or within 48 hours after admission.

### Results

400 children were included in the study and of these, 58% were males. Most (87%) of the children were aged 24 months and below. The prevalence of pneumonia was 28%

Protection and Privacy and the European act on General Data Protection Regulation do not allow for personal data to be made available to other researchers without prior written approval from relevant institutions and authorities. The Data Protection Officer of the University of Copenhagen can be contacted about data inquiries at dpo@adm. ku.dk".

**Funding:** In this study we used secondary data obtained from ProbiSAM trial that was funded by Chr. Hansen A/S, University of Copenhagen and Innovation Fund Denmark. I was granted permition to utilize the data by principal investigator of the ProbiSAM trial. So the ProbiSAM trial team did the data collection. However, they had no role played in the study design, analysis, preparation of the manuscript and decision to publish. Apart from using secondary data from ProbiSAM trial the study was unfunded. The authors received no specific funds.

**Competing interests:** The authors have declared no competing interests.

(95% CI: 24–33). In multivariable analysis, factors associated with prevalence of pneumonia include: female sex (Adjusted PR: 1.72, 95% CI: 1.20–2.45), stunting (Adjusted PR: 1.60, 95% CI: 1.12–-2.30) and presumptive TB (Adjusted PR: 1.95, 95% CI: 1.34–-2.82). Incident pneumonia was estimated at 356 (95% CI: 305–-416) per 1000 hospital admissions of children with SAM. In multivariable analysis, children at risk of pneumonia were: young age 6 to 12 months (Adjusted RR: 1.89, 95% CI: 1.10–3.29), stunting (Adjusted RR: 1.59, 95% CI: 1.15–2.20), HIV infection (Adjusted RR: 1.59, 95% CI: 1.08–2.33), presumptive TB (Adjusted RR: 1.73, 95% CI: 1.23–2.43), used a nasogastric tube for feeding (RR: 1.38, 95% CI: 1.00–1.89) and prolonged hospitalization of 15 days or more (Adjusted RR: 2.16, 95% CI: 1.39–-3.35).

### Conclusion and Significance

This study identified that prevalence of pneumonia among children hospitalized with SAM is still high and similar to previous studies done in different settings in Uganda. In addition, this study showed that the risk of incident pneumonia among children hospitalized with SAM is high. These findings point out the need to develop guidelines to monitor, detect and treat incident pneumonia among children hospitalized with SAM. Stunting as factor associated with both prevalence and incidence of pneumonia underscores the importance of further research to evaluate how stunting relates to pneumonia.

## Introduction

Pneumonia is the number one infectious killer of children aged less than 5 years globally [1]. Current evidence shows that pneumonia claims the lives of over 700,000 children under the age of 5 years every year [1]. The highest proportion of these deaths are in sub-Saharan Africa (SSA) and Asia [1].

Malnutrition is an important risk factor for pneumonia and has been shown to increase the frequency and severity of pneumonia episodes [2]. According to the global estimates for child malnutrition, 13.7 million children less than 5 years are severely wasted and the majority are from low income countries (LICs) [3]. According to the Uganda demographic health survey (UDHS) 2022, the prevalence of wasting among children under the age of 5 years is 3.2% [4].

Pneumonia is still a big burden in children hospitalized with severe acute malnutrition (SAM) and it is associated with poor outcomes including death and prolonged hospital stay [5,6]. Over half of in-hospital child pneumonia deaths in low-income and middle-income countries are attributable to malnutrition [5]. A systematic review by Beletew et al, indicated that the prevalence of pneumonia in Eastern Africa is still high (34%) [7]. According to the weekly surveillance data for 2022 from Ministry of Health in Uganda, the incidence of severe pneumonia was 108 per 100,000 children under five years. Similarly, the prevalence of pneumonia in children with SAM is high ranging from 17% to nearly 40% in different hospital settings [8–10] and is associated with up to four-fold risk of mortality [5].

Most children with pneumonia are reported to have community-acquired infections. However, children with SAM are at a higher risk of developing hospital-acquired infections like pneumonia given their immunocompromised state and prolonged hospital stay [11,12]. And yet, children with SAM are hospitalized for long and coupled with their low immunity makes them more prone to acquiring incident pneumonia during hospitalization. Furthermore, incident pneumonia acquired during hospitalization is reported to be caused by multidrug resistant organisms that are associated with high mortality and prolonged hospital stay leading

to increased hospital-related costs in terms of drugs, monitoring by doctors/nurses and food [12]. Despite the poor outcomes related with incident pneumonia acquired during hospitalization, there are hardly any studies describing its magnitude among children with SAM in our context.

Studies have reported factors associated with pneumonia among children less than 5 years to include: younger age, use of wood as a fuel source, not up-to-date for age immunization status, severe acute malnutrition, lack of exclusive breast feeding, rural residence, exposure to cigarette smoking and cooking food in the living room [7,13,14]. Similarly, studies have documented malnutrition, prolonged hospital stay and low immunity as risk factors for developing incident pneumonia during hospitalization [12,15,16]. However, there is limited data on the characteristics of severely malnourished children who present with pneumonia at admission or acquire incident pneumonia during hospitalization. Therefore, in this study, we aimed to determine the prevalence, incidence and associated factors, of pneumonia among children hospitalized with SAM at the nutrition unit of a tertiary care hospital in Uganda. The information generated from this study may be used to inform policy and help clinicians develop guidelines to monitor, detect and treat pneumonia among children hospitalized with SAM. This will eventually improve care of severely malnourished children hospitalized with pneumonia and reduce on the mortality associated with it.

## Materials and methods

### Study design and setting

This study employed 2 study designs; a) cross sectional design to determine the prevalence of pneumonia and describe associated factors and b) single arm retrospective cohort design to determine the incidence of pneumonia during hospitalization for children with SAM and describe associated factors. In this study we used secondary data from a randomized controlled trial (ProbiSAM) that assessed the effect of Probiotics on diarrhea in children hospitalized with SAM (www.isrctn.com with registration number ISRCTN16454889 [17]. Details of the ProbiSAM trial are reported elsewhere [18]. The ProbiSAM trial was conducted at Mwanamugimu Nutritional Unit, Mulago National Referral and Teaching Hospital between 10th March 2014 to 8th July 2015. Mulago hospital is located in the Central region of Uganda and it serves a number of urban and rural districts surrounding central Uganda and all referral cases from all over the country. Mwanamugimu Nutrition Unit (MNU) is one of the specialized unit in Mulago Hospital. The nutrition unit attends all malnourished children and comprises the stabilization and rehabilitation wards, as well as an outpatient clinic. During inpatient care, children are stabilized by treating all the medical complications according to the Uganda national protocol "Integrated Management of Acute Malnutrition (IMAM) [19] and then transferred to a rehabilitation ward where they are prepared for discharge into outpatient therapeutic clinic (OTC) for follow up. The average monthly attendance for inpatients is 120 children aged 1 month to 12 years, 75% of whom are aged 6 to 59 months.

### Study procedures

### Study population

The study population included children aged 6 to 59 months who had SAM and were enrolled in the ProbiSAM trial. Patients with severe conditions like shock or severe respiratory distress at admission, weight below 4.0 kg, obvious disability or significant congenital or malignant disease and patients admitted with SAM the previous 6 months were excluded.

**Sample size estimation.** To determine the prevalence of pneumonia and associated factors, we calculated the sample size basing on Ugandan studies done in Mwanamugimu-Kampala, Mbarara and Jinja on prevalence of pneumonia in children with SAM at an average of 26% [8–10]. Using Kish Leslie formula for cross sectional studies [20] the minimum sample size required to determine the prevalence of pneumonia was 301 children. Adjusting for 10% non-response rate, the minimum sample size was 332 children. However, in the current paper we utilized all the 400 study participants from the randomized trial, therefore our study was powered enough to make an inference. To determine the incidence of pneumonia and its associated factors, we estimated a sample size using formula for independent cohort studies [21]. We computed sample size basing on a study by Olivier et al who reported the prevalence of hospital acquired pneumonia to be 33% [22]. Considering the power at 80%, 5% level of significance, RR=1.5 and 1 control per exposed subject, the required sample was 215 per group. The study utilized the single-arm retrospective cohort (only exposed group), therefore the minimum sample size was 215 subjects. Since we utilized the single-arm retrospective cohort (only exposed group), the number of children that were at risk of acquiring pneumonia during hospitalization were 289.Therefore our study was powered enough to make an inference.

**Study variables.** We extracted data from the ProbiSAM trial data base. The trial enrolled 400 children aged between 6 to 59 months hospitalized with SAM. The children were randomized into a treatment arm to receive a daily dose of probiotics (Lactobacillus and Bifidobacterium) and placebo arm to receive a dummy sachet of similar appearance and taste like in the other arm, from admission up to discharge. Each arm had 200 children. Data were collected using a validated assessment tool at the time of admission and during follow-up.

**At admission.** At admission, trained research assistants that included pediatrician, medical officer, nurses and nutritionist collected data according to their area of specialization.The pediatrician or medical officer collected data on demographics that included age, sex and clinical characteristics of cough or difficulty in breathing, respiratory rate, chest-indrawing, auscultatory findings in the chest, oxygen saturation, grunting, and general danger signs (inability to breastfeed or drink, lethargy or unconscious and convulsions). Diagnosis of pneumonia was made according to the WHO clinical criterion for pneumonia [23]. The clinicians also collected additional clinical characteristics of presence of oedema, fever, temperature, diarrhoea, vomiting, history of any illness in the 2 weeks prior to enrolment and loss of appetite. Anthropometric measurements were done by nutritionists and diagnosis of SAM made according to WHO criteria [24]. Length/height was measured using an infant length board (infant/child shorr-board, Maryland, USA) and MUAC was assessed using a measuring tape, both to the nearest 0.1cm. Bodyweight was measured using a digital scale (Seca 813, Hamburg, Germany) to the nearest 0.1 kg. Z-scores were calculated using the WHO child growth standards [25]. The nurses collected information on the social characteristics that included: house hold size, breastfeeding status, immunisation status and caregiver's level of education. A visual analogue scale (VAS) assessing the perceived severity of the child's illness by the caretaker was filled in at admission and daily during inpatient therapeutic care (ITC). The VAS had faces with facial expressions of children and scores ranging between 0 and 10 according to how sick the caretaker thought the child was. Blood samples were taken off to do laboratory measurements that included: HIV serostatus, C-reactive protein (CRP), white blood cell count and a blood smear for malaria parasites. The mother's HIV status was also done to determine HIV-exposure status of their children. Chest radiographs were done for children who had pneumonia at admission or had presumptive TB. The details of how the different procedures were done in the ProbiSAM trial have been reported elsewhere [18]. We extracted all the variables collected at admission as indicated above from the ProbiSAM data set and

analysed them to determine the prevalence and associated factors of pneumonia which is the first part of the objective of our study.

**During hospitalization.** During hospitalisation, the child was evaluated for clinical symptoms and signs of pneumonia on daily basis until discharge from the study. The clinical signs included cough or difficulty in breathing, respiratory rate, chest-in drawing and oxygen saturation. Diagnosis of pneumonia was made as per the WHO clinical criterion. Date of onset and stop of these clinical signs of pneumonia were indicated in the questionnaire. If the pneumonia episode reappeared some days after admission in the same child, it was considered part of the previous pneumonia episode. Other clinical characteristics that were collected during hospitalization include degree of oedema, pulse rate, dehydration, fever, temperature, use of a nasogastric tube for feeding, use of Resomal for dehydration, choice of antibiotics given, diagnosis of malaria and date of discharge or death to determine duration of hospitalization. The nutritionist took daily weight and other anthropometric measurements (height, sitting length, MUAC) were taken weekly during hospitalization. Children were treated according to the Uganda national protocol "Integrated Management of Acute Malnutrition (IMAM) [19].

Children who were diagnosed with pneumonia at admission or during hospitalization received intravenous antibiotics of ampicillin and gentamycin as the first line treatment. Those that failed to respond within 48 to 72 hours and those with severe pneumonia were given intravenous ceftriaxone. All children received antibiotics over a period ranging from 5 to 10 days. Re-feeding was started with F-75 formula (Nutriset, Malaunay, France) every 2 hours at 100 -130 ml/kg/day depending on presence or absence of oedema according to the National Protocol for management of malnutrition [19]. When children were unable to feed orally, feeds were given via a nasogastric tube. When oedema plus medical complications resolved and appetite returned, children had an acceptance test performed to evaluate if they can consume a predefined amount of RUTF. When they passed the acceptance test, they were gradually transitioned to RUTF over 2-3 days. We extracted additional variables collected during hospitalization from the data set and analysed them to determine the incidence and risk factors of pneumonia which is the second part of the objective of our study.

## Operational definitions

Severe acute malnutrition: was when a the child had a mid-upper arm circumference (MUAC) of less than 11.5cm or weight-for-height Z score below minus three standard deviations from the median of the WHO child growth standards (WHZ < -3SD Z score) and or the presence of bipedal pitting oedema [24].

Pneumonia was defined as presence of cough or difficulty in breathing and age-specific fast breathing (respiratory rate of ≥50 breaths per minute for those below 12 months and ≥40 breaths per minute for those above ≥12 months) or chest in-drawing according to world health organisation (WHO) criterion [23].

Severe pneumonia was defined as presence of signs and symptoms of pneumonia as indicated above plus at least one of the following: central cyanosis or oxygen saturation less than 90% on pulse oximetry, very severe respiratory distress (with grunting and very severe chest in drawing), a general danger sign (inability to breastfeed or drink, lethargy or unconscious, convulsions) [23].

Prevalence of pneumonia (pneumonia at admission): was defined as proportion of children diagnosed as cases of pneumonia at the time of enrolment into the study or within the first 48 hours of admission.

Incidence of pneumonia (pneumonia acquired during hospitalization): was defined as proportion of children who developed clinical signs of pneumonia for the first time during hospitalization among children who did not have a diagnosis of pneumonia at admission or within 48 hours after admission.

## Statistical analysis

Data were analyzed using STATA version.17 (Stata Corp, College Station, TX, 2015). Descriptive analysis included frequencies (proportions) for categorical variables and means (Standard deviations-SD) for normally distributed variables or medians (Interquartile range) for non-normally distributed variables. We compared study participant characteristics stratified by the outcome (with or without pneumonia) at admission and during hospitalization using Chi-square test statistics for categorical variables and T-test statistics for continuous variables. We fitted two models for prevalence and incidence of pneumonia.

Model 1: Generalised linear models (GLM) were fitted to the cross-sectional dataset describing the prevalence of pneumonia at admission, using maximum likelihood estimation. Due to the binary outcome variable for pneumonia, a binomial likelihood with a log link function was used to estimate the prevalence and perform univariate and multivariate regression of predictors of pneumonia at admission. Crude and adjusted prevalence ratios (CPR & APR) were estimated with the corresponding 95% confidence intervals.

Model 2: Generalised linear models were fitted to the cohort dataset describing the incidence of pneumonia during hospitalisation, using maximum likelihood estimation. A Poisson likelihood with a log link function was used to estimate the incidence of pneumonia and perform univariate and multivariable regression modelling of risk factors. The crude and adjusted relative risks (RRs) with corresponding 95% confidence interval were reported.

Model fitting was carried out in steps; 1) fitted only intercepts model, 2) full model included all variables that had a p-value <0.20 at bivariate analysis or known risk factors, and 3) a reduced model. In both models', we tested for collinearity using variance inflation factors (VIF) and if two or more variables were found to be related, one was dropped. We further, tested for interaction between different variables during the analysis. The variables tested for interaction include; oedema and age, oedema and weight for height Z score, fever and age. Interaction was only present in fever and age. Model goodness of fit were evaluated using Akaike Information Criterion (AIC) and Bayesian Information criterion (BIC).

## Missing data

In the ProbiSAM data set, we found some variables of interest in our study had missing data particularly HIV status of the baby and mother, C-reactive protein (CRP)and White blood cell count. We did sensitivity analysis for this variables that had missing data. We ran a model with or without missing data for each particular variable and compared differences in the results. In addition, we also compared if the models with or without missing data would affect the estimated sample size and hence power of our study. Since we found no differences in the findings, we reported findings of those with complete data.

## Ethical approval

Our study utilized secondary data from ProbiSAM trail. Ethical approval for the ProbiSAM trial was obtained from Makerere University School of Medicine Research and Ethics Committee (Reference number: REC REF 2013-132) with a consultative approval from the National Committee of Health Research Ethics in Denmark. The ProbiSAM trial was also

approved by Uganda National Council for Science and Technology (Reference number: HS 1503) a regulatory body which gives oversight to all scientific research in Uganda. A written informed consent or accent for minors was obtained from all care givers before enrolment into the ProbiSAM trial. All methods were performed in accordance with the relevant guidelines and regulations.

## Results

### Prevalence of pneumonia and associated factors

**Characteristics of severely malnourished children at admission.** There were 400 children in this study, 58% were males. Most (87%) of the children were aged 24 months and below. Seventy-five percent (75%) had received the required immunization doses for that age (Table 1). Two thirds of the children in this study had oedema. About half of the children had severe wasting and half had severe stunting. Only 15% of the children were found to still be breastfeeding among those less than 2 years. Eleven percent (11%) of the children had confirmed HIV infection and 18% were HIV-exposed. About half of the children had mothers/caregivers with no formal education.

Chest radiographs were done for 129 children who had pneumonia at admission or had presumptive TB. Of these 70% (90/129) had pathological findings (infiltrates or consolidation or pleural effusion). Among those who demonstrated clinical signs of pneumonia at admission, 85% (51/60) had radiological findings confirming presence of pneumonia (Details of these results are not shown in the Table 1).

Prevalence of pneumonia among children admitted with SAM.

The prevalence of pneumonia at admission was found to be 28% (95% CI: 24–33) among children admitted with SAM.

**Factors associated with prevalence of pneumonia.** Female Sex, presence of bipedal oedema, WHZ, HAZ, MUAC, CRP, presumptive TB and HIV status of the mother were factors associated with presence of pneumonia at admission on bivariate analysis (Table 2). The prevalence of pneumonia at admission was 65% higher among female children compared to males (cPR: 1.65; 95% CI: 1.20–2.27). Children with oedema, the prevalence of pneumonia at admission was 35% lower compared to children without oedema at admission (cPR: 0.65; 95% CI: 0.48–0.89).

In the multivariable regression model, female Sex, HAZ and presumptive TB were factors associated with pneumonia at admission (Table 2). The prevalence of pneumonia at admission was 72% higher among female children compared to males (Adjusted PR: 1.72, 95% CI: 1.20–2.45). Further, the prevalence of pneumonia at admission was 60% higher among children with a HAZ score of < -3SD compared to those with a Z score of $\geq$ -3 SD (Adjusted PR: 1.60, 95% CI: 1.12–2.30). Children with presumptive TB, the prevalence of pneumonia at admission was 95% higher compared to those with no presumptive TB (Adjusted PR: 1.95, 95% CI: 1.34–2.82).

### Incidence of pneumonia and associated factors

**Characteristics of severely malnourished children at risk of acquiring pneumonia during hospitalization.** Of the 289 children that did not have pneumonia at admission, 36% (103/289) developed signs and symptoms of pneumonia (cough or difficulty in breathing, fast breathing, chest in drawing) (Table 3). Most (87%) of the children were aged 24 months and below. About half of the children had axillary temperature recording of more than 37.5$^0$C

**Table 1. Baseline characteristics of severely malnourished children at admission.**

| Variable | N | Frequency n (%) |
|---|---|---|
| **Demographics** | | |
| Median age (months) | 400 | |
| 6–12 | | 158 (39) |
| 13–24 | | 191 (48) |
| 25–60 | | 51 (13) |
| Sex | 400 | |
| Male | | 230 (58) |
| Female | | 170 (42) |
| Median household size[t] | 385 | 5 (3–6) |
| Immunization status | 389 | |
| Not up- to -date | | 96 (25) |
| Up-to-date[a] | | 293 (75) |
| **Nutritional *status*** | | |
| Oedema | 400 | |
| Absent | | 139 (35) |
| Present | | 261 (65) |
| WHZ | 387 | |
| < -3 SD | | 178 (46) |
| ≥ -3 SD | | 209 (54) |
| HAZ | 387 | |
| < -3 SD | | 201 (52.0) |
| ≥ -3 SD | | 186 (48.0) |
| MUAC (cm) | 399 | |
| < 11.5 | | 206 (52) |
| ≥ 11.5 | | 193 (48) |
| Breastfeeding status[b] | 349 | |
| Stopped BF | | 272 (78) |
| Still BF | | 54 (15) |
| Unknown | | 23 (7) |
| Birth weight (kg) | 271 | |
| < 2.5 | | 30 (11) |
| ≥ 2.5 | | 241 (89) |
| **Medical history at admission** | | |
| Fever | 399 | |
| No | | 188 (47) |
| Yes | | 211 (53) |
| Diarrhea | 399 | |
| No | | 155 (39) |
| Yes | | 244 (61) |
| Severity of sickness[c] | 399 | |
| 0–4 | | 60 (15) |
| 5–10 | | 339 (85) |
| **Physical examination** | | |
| Oral thrush | 399 | |
| No | | 315 (79) |
| Yes | | 84 (21) |
| Dermatosis | 400 | |
| No | | 373 (93) |
| Yes | | 27 (7) |
| Temperature | 400 | |
| < 37.5 °C | | 318 (79) |
| ≥ 37.5 °C | | 82 (21) |
| **HIV *status*** | 368 | |
| Exposed | | 72 (18) |
| Negative | | 253 (63) |
| Positive | | 43 (11) |

(*Continued*)

**Table 1**. (Continued)

| Variable | N | Frequency n (% ) |
|---|---|---|
| **Laboratory characteristics** | | |
| C-reactive protein (CRP) | 352 | |
| < 10 mg/l | | 134 (38) |
| $\geq$ 10 mg/L | | 218 (62) |
| Haemoglobin[t] | 298 | 8.78 ($\pm$2.18) |
| White blood cell count | 298 | |
| $\leq$ 12 x10$^9$/l | | 173 (58) |
| > 12 x10$^9$/l | | 125 (42) |
| **Comorbidities[d]** | | |
| Tuberculosis | 400 | |
| Not Presumptive | | 324 (81) |
| Presumptive | | 76 (19) |
| Septicemia | 398 | |
| Not Suspected | | 302 (76) |
| Suspected | | 96 (24) |
| **Caregiver characteristics** | | |
| **Education Level of mother** | 389 | |
| Not completed primary education | | 176 (45) |
| Completed primary education | | 157 (40) |
| Secondary + | | 56 (14) |
| **HIV status of mother** | 400 | |
| Positive | | 83 (21) |
| Negative | | 251 (63) |
| Unknown | | 66 (16) |

Abbreviations: WHZ, weight for height z score, HAZ, height for age Z score, MUAC, mid-upper arm circumference, IQR, interquartile range, BF, breastfeeding

[a] Immunisation status up-to-date: have completed receiving the recommended PCV 3

[b] Breastfeeding status: we considered children less than 2 years who were expected to still be breastfeeding by WHO standards

[c] Severity of Sickness is evaluated by the child's caregiver on a Visual Analogue Scale (VAS) from 0 to 10.

[d] Comorbidities: a disease or medical condition that is simultaneously present in severely malnourished children at admission.

during hospitalization. A third (36%) had difficulty in feeding and were fed through a naso-gastric tube (NGT). Two thirds of the children (65%) spent 15 days or more during hospitalization. All other characteristics of the study participants were similar both at admission and during hospitalization.

Sixty-nine (69) children had chest radiographs done and out of these 57% (39/69) had pathological findings (infiltrates or consolidation or pleural effusion) suggestive of pneumonia. Among those who demonstrated clinical signs of pneumonia during hospitalization,70% (35/50) had radiological findings confirming presence of pneumonia (details of the results not shown in the Table 3).

**Incidence of pneumonia among children hospitalized with SAM.** The incidence of pneumonia was estimated at 356 (95% CI: 305–416) per 1000 hospital admissions of children with SAM. In the stratified analysis, the incidence of pneumonia was higher among male children 361 (95% CI: 297–439) per 1000 hospital admissions with SAM compared to females 349 (95% CI: 269–451) per 1000. However, the difference was not statistically significant (p value 0.831).

## Factors associated with incidence of pneumonia during hospitalization

Bipedal edema, WHZ, HAZ, MUAC, breastfeeding, HIV positive status, presumptive TB, use of NGT to feed and prolonged hospitalization were factors associated with incidence

Table 2. Factors associated with prevalence of pneumonia among children with SAM.

| Variable | Pneumonia status at admission[a] | Pneumonia status at admission[a] | Crude PR | Adjusted PR |
|---|---|---|---|---|
| | Pneumonia (N = 111) | No Pneumonia (N = 289) | | |
| | n (%) | n (%) | | |
| **Variable** | | | | |
| **Demographics** | | | | |
| Median age (months) | | | | |
| 6–12 | 45 (40) | 113 (39) | 1.12 (0.65–1.90) | 1.44 (0.82–2.53) |
| 13–24 | 53 (48) | 138 (48) | 1.09 (0.64–1.84) | 1. 10 (0.63–1.92) |
| 25–60 | 13 (12) | 38 (13) | 1 | 1 |
| Sex | | | | |
| Male | 50 (45.0) | 180 (62.0) | 1 | 1 |
| Female | 61 (55) | 109 (38) | **1.65 (1.20–2.27)**** | **1.72 (1.20–2.45)**** |
| Median household size[t] | 5 (3–6) | 4 (3–6) | | |
| Immunization status | | | | |
| Up- to -date [b] | 80 (74) | 213 (76) | 1 | |
| Not-up-to-date | 28 (26) | 68 (24) | 1.06 (0.75–1.50) | |
| **Nutritional *status*** | | | | |
| Oedema | | | | |
| Absent | 50 (45) | 89 (31) | 1 | 1 |
| Present | 61 (55) | 200 (69) | **0.65 (0.48–0.89)**** | 0. 85 (0.57–1.26) |
| WHZ | | | | |
| < -3 SD | 63 (58) | 115 (41) | **1.64 (1.18–2.28)**** | |
| ≥ -3 SD | 45 (42) | 164 (59) | 1 | |
| HAZ | | | | |
| < -3 SD | 67 (62) | 134 (48) | **1.51 (1.08–2.11)**** | **1.60 (1.12–2.30)**** |
| ≥ -3 SD | 41 (38) | 145 (52) | 1 | |
| MUAC (cm) | | | | |
| < 11.5 | 72 (65) | 134 (47) | **1.72 (1.23–2.42)**** | |
| ≥ 11.5 | 39 (35) | 154 (54) | 1 | |
| Breastfeeding status[c] | | | | |
| Stopped BF | 70 (71) | 202 (80) | 0.69 (0.46–1.04) | |
| Still BF | 20 (20) | 34 (14) | 1 | |
| Unknown | 8 (8) | 15 (6) | 0.94 (0.49–1.81) | |
| Birth weight (kg) | | | | |
| < 2.5 | 10 (14) | 20 (10) | 1 | |
| ≥ 2.5 | 62 (86) | 179 (90) | 1.29 (0.74–2.25) | |
| **Medical history at admission** | | | | |
| Diarrhea | | | | |
| No | 41 (37) | 114 (40) | | |
| yes | 70 (63) | 174 (60) | 1.08 (0.78–1.50) | |
| Severity of sickness [d] | | | | |
| 0–4 | 12 (11) | 48 (17) | 1 | |
| 5–10 | 99 (89) | 240 (83) | 1.46 (0.85–2.48) | |
| **Physical examination** | | | | |
| Oral thrush | | | | |
| No | 84 (76) | 231 (80) | | |
| yes | 26 (24) | 58 (20) | 1.16 (0.80–1.68) | |
| Dermatosis | | | | |
| No | 105 (95) | 268 (93) | | |
| yes | 6 (5) | 21 (7) | 0.79 (0.382–1.63) | |
| **Lab characteristics** | | | | |
| C-reactive protein (CRP) | | | | |
| < 10 mg/L | 26 (27) | 108 (42) | 1 | 1 |

(*Continued*)

**Table 2.** (Continued)

| Variable | Pneumonia status at admission[a]  Pneumonia  (N = 111) | Pneumonia status at admission[a]  No Pneumonia  (N = 289) | Crude PR | Adjusted PR |
|---|---|---|---|---|
| | n (%) | n (%) | | |
| ≥ 10 mg/L | 71 (73) | 147 (58) | **1.68 (1.13–2.49)**\*\* | 1.44 (0.95–2.19) |
| Hemoglobin | 8.79 (±2.50) | 8.80 (±2.05) | 0.98 (0.89–1.09) | |
| White blood cell count | | | | |
| < 12 x$10^9$/l | 43 (50) | 130 (61) | 1 | |
| ≥ 12 x$10^9$/l | 43 (50) | 82 (39) | 1.38 (0.97–1.97) | |
| HIV status | | | | |
| Exposed | 23 (21) | 49 (17) | 1.26 (0.84–1.88) | 1.10 (0.71–1.72) |
| Negative | 64 (58) | 189 (65) | 1 | 1 |
| Positive | 15 (13) | 28 (10) | 1.38 (0.87–2.18) | 1.04 (0.58–1.88) |
| **Comorbidities[e]** | | | | |
| Tuberculosis | | | | |
| Not presumptive | 76 (68) | 248 (86) | 1 | 1 |
| Presumptive | 35 (32) | 41 (14) | **1.96 (1.43–2.68)**\*\*\* | **1.95 (1.34–2.82)**\*\*\* |
| Septicemia | | | | |
| Not suspected | 79 (72) | 223 (77) | | |
| Suspected | 31 (28) | 65 (23) | 1.23 (0.87–1.74) | |
| **Maternal factors** | | | | |
| Mothers HIV status | | | | |
| Positive | 27 (24) | 56 (19) | 1.38 (0.94–2.03) | |
| Negative | 59 (53) | 192 (66) | 1 | |
| Unknown | 25 (22) | 41 (14) | **1.61(1.10–2.36)**\*\* | |

[t]T-test statistic, \*P<0.1, \*\*p<0.05, \*\*\*p<0.001

Abbreviations: WHZ, weight for height z score, HAZ, height for age Z score, MUAC, mid-upper arm circumference, IQR, interquartile range, BF, breastfeeding.

[a] proportions were compared using $\chi^2$ test and continuous variables were compared using [t]T-test statistic.

[b] Immunisation status up-to-date: have completed receiving the recommended PCV 3.

[c] Breastfeeding status: we considered children less than 2 years who expected to still be breastfeeding by WHO standards.

[d] Severity of sickness is evaluated by the child's caregiver on a Visual Analogue Scale (VAS) from 0 to 10.

[e] Comorbidities: a disease or medical condition that is simultaneously present in severely malnourished children who are the participants.

of pneumonia on bivariate analysis (Table 4). Children with oedema were 30% less likely to acquire pneumonia during hospitalization compared to those without oedema at admission (RR: 0.70, 95% CI: 0.51–0.95). Children with WHZ of < -3SD, were 43% more likely to develop pneumonia during hospitalization compared to those with a WHZ score of ≥ -3 SD (RR: 1.43, 95% CI:1.04–1.94). Children who used a nasogastric tube (NGT) for feeding during hospitalization were 54% more likely to acquire pneumonia compared to those without an NGT (RR: 1.54, 95% CI: 1.14–2.09).

In the multivariable regression model, age, HAZ, HIV positive status, presumptive TB, use of a NGT for feeding and prolonged hospitalization were risk factors for incident pneumonia during hospitalization. The younger children aged between 6 to 12 months, were 89% more likely to develop pneumonia during hospitalization (Adjusted RR: 1.89, 95% CI: 1.10–3.29). Children with HAZ of < -3SD were 59% more likely to develop pneumonia during hospitalization compared to those with HAZ of ≥ -3 SD (Adjusted RR: 1.59, 95% CI: 1.15–2.20). HIV positive children, were 59% more likely to develop pneumonia during hospitalization compared to those who were HIV negative (Adjusted RR: 1.59, 95% CI: 1.08–2.33). Children with presumptive TB were 73% more likely to develop pneumonia during hospitalization compared to those with no presumptive TB (Adjusted RR: 1.73, 95% CI: 1.23–2.43). Children who

**Table 3. Characteristics of severely malnourished children who were at risk of acquiring pneumonia during hospitalization (N = 289).**

| Variable | N | Frequency n (%) |
|---|---|---|
| **Demographics** | | |
| Age (months) | 289 | |
| 6–12 | | 113 (39) |
| 13–24 | | 138 (48) |
| 25–60 | | 38 (13) |
| Sex | 289 | |
| Male | | 180 (62) |
| Female | | 109 (38) |
| Immunization status | 289 | |
| Not up- to -date | | 76 (26) |
| Up-to-date[a] | | 213 (74) |
| **Nutritional *status*** | | |
| Oedema | 289 | |
| Absent | | 89 (31) |
| Present | | 200 (69) |
| WHZ | 279 | |
| < -3 SD | | 115 (41) |
| ≥ -3 SD | | 164 (59) |
| HAZ | 279 | |
| < -3 SD | | 134 (48) |
| ≥ -3 SD | | 145 (52) |
| MUAC (cm) | 288 | |
| < 11.5 | | 134 (52) |
| ≥ 11.5 | | 154 (48) |
| Breastfeeding status[b] | 251 | |
| Stopped BF | | 202 (80) |
| Still BF | | 34 (14) |
| Unknown | | 15 (6) |
| Birth weight (kg) | 199 | |
| < 2.5 | | 20 (10) |
| ≥ 2.5 | | 179 (90) |
| **Laboratory characteristics at admission** | | |
| C-reactive protein (CRP) | 255 | |
| < 10 mg/l | | 108 (42) |
| ≥ 10 mg/L | | 147 (58) |
| White blood cell count | 212 | |
| ≤ 12 x$10^9$/l | | 130 (61) |
| > 12 x$10^9$/l | | 82 (39) |
| HIV status | 266 | |
| Exposed | | 49 (18) |
| Negative | | 189 (71) |
| Positive | | 28 (10) |
| Tested positive for malaria | 104 | |
| No | | 87 (84) |
| Yes | | 17 (16) |
| **Comorbidities[c]** | | |
| Tuberculosis | 289 | |
| Not Presumptive | | 248 (85) |
| Presumptive | | 41 (14) |
| **Clinical characteristics during hospitalization** | | |
| Temperature | 289 | |
| < 37.5 ∘C | | 144 (50) |
| ≥ 37.5 ∘C | | 145 (50) |
| Pulse | 273 | |
| Normal | | 250 (92) |

(*Continued*)

**Table 3.** (Continued)

| Variable | N | Frequency n (% ) |
|---|---|---|
| Abnormal | | 23 (8) |
| Use of Resomal | 289 | |
| No | | 61 (21) |
| Yes | | 228 (79) |
| Use of Nasogastric tube | 285 | |
| No | | 182 (64) |
| Yes | | 103 (36) |
| Duration of hospital stay | 289 | |
| 0 to 14 days | | 104 (36) |
| 15 days or more | | 184 (65) |

Abbreviations: WHZ, weight for height z score, HAZ, height for age Z score,
MUAC, mid-upper arm circumference, IQR, interquartile range, BF, breastfeeding
[a] Immunisation status up-to-date: have completed receiving the recommended PCV 3
[b] Breastfeeding status: we considered children less than 2 years who expected to still be breastfeeding by WHO standards
[c] Comorbidities: a disease or medical condition that is simultaneously present in severely malnourished children who are the participants.

used an NGT during hospitalization were 38% more likely to develop pneumonia compared to those without an NGT (RR: 1.38, 95% CI: 1.00–1.89). Children who had prolonged hospitalization of 15 days or more, were 2 times more likely to develop pneumonia (Adjusted RR: 2.16, 95% CI: 1.39–3.35).

## Discussion

In this study, we determined the prevalence, incidence and associated factors of pneumonia among children presenting with SAM at the nutrition unit of a tertiary care hospital in Uganda.

### Prevalence of pneumonia and associated factors

**Prevalence of pneumonia.** The prevalence of pneumonia was 28%. The findings in the current study are consistent with previous studies done in Uganda and other developing countries where the prevalence of pneumonia among children with SAM ranged from 20 to 40% [9,10,26,27]. However, the prevalence of 28% is still very high and similar to the current findings of pneumonia among malnourished children in our settings in Jinja (37.6%) and Mbarara (26.8%) [9,10]. And yet to effectively prevent, protect and treat against pneumonia by the year 2025, a multifaceted approach was needed with special focus on vaccination, improved hygiene, adequate nutrition, and access to timely and appropriate medical care [28]. Efforts have been made to address some of this approaches however with severe acute malnutrition we still have a big burden of 3.2 % according UDHS 2022 [4]. Therefore, there is urgent need to address malnutrition to contribute to reducing the high burden of pneumonia in children. In contrast, the prevalence of 28% in the current study is much lower than the one reported in another study done in Bangladesh where they found a prevalence of 68% [29]. The difference in prevalence between these two studies could be explained by the method used in making diagnosis of pneumonia. In the current study, we based on WHO clinical criteria to make a diagnosis of pneumonia compared to Bangladesh where they used both clinical and radiological criteria [29]. In the current study, CXR were done to confirm diagnosis for only children who presented with clinical symptoms of pneumonia. And yet, evidence shows that clinical signs are poor predictors of pneumonia in severely malnourished children [30]. There is a possibility that some children with pneumonia were missed at admission making

**Table 4. Factors associated with incidence of pneumonia among severely malnourished children during hospitalization.**

| | Pneumonia status during hospitalization[a] | Pneumonia status during hospitalization[a] | Crude PR | Adjusted PR |
|---|---|---|---|---|
| **Variable** | Pneumonia (N = 103) | No Pneumonia (N = 186) | | |
| **Variable** | n (%) | n (%) | | |
| **Demographics** | | | | |
| Age (months) | | | | |
| 6–12 | 46 (45) | 67 (36) | 1.41 (0.81–2.42) | **1.89 (1.10–3.29)**\*\* |
| 13–24 | 46 (45) | 92 (49) | 1.15 (0.66–2.00) | 1.23 (0.71–2.13) |
| 25–60 | 11 (10) | 27 (14) | 1 | 1 |
| Sex | | | | |
| Male | 65 (63) | 115 (62) | 1 | |
| Female | 38 (37) | 71 (38) | 0.97 (0.69–1.33) | – |
| Immunization status | | | | |
| Not up- to -date | 30 (29) | 46 (25) | 1.15 (0.82–1.60) | – |
| Up-to-date[b] | 73 (71) | 140 (75) | 1 | |
| **Nutritional *status*** | | | | |
| Oedema | | | | |
| Absent | 40 (39) | 49 (26) | 1 | |
| Present | 63 (61) | 137 (74) | **0.70 (0.51–0.95)**\*\* | – |
| WHZ | | | | |
| < -3 SD | 50 (50) | 65 (36) | **1.43 (1.04–1.94)**\*\* | – |
| ≥ -3 SD | 50 (50) | 114 (64) | 1 | |
| HAZ | | | | |
| < -3 SD | 56 (56) | 78 (44) | **1.38 (1.00–1.89)**\*\* | **1.59 (1.15–2.20)**\*\* |
| ≥ -3 SD | 44 (44) | 101 (56) | 1 | 1 |
| MUAC (cm) | | | | |
| < 11.5 | 61 (59) | 73 (39) | **1.68 (1.21–2.29)**\*\* | – |
| ≥ 11.5 | 42 (41) | 112 (60) | 1 | |
| Breastfeeding status[c] | | | | |
| Stopped BF | 65 (71) | 137 (86) | **0.48 (0.35–0.64)**\*\*\* | – |
| Still BF | 23 (25) | 11 (7) | 1 | |
| Unknown | 4 (4) | 11 (7) | **0.39 (0.16–0.94)**\*\* | – |
| Birth weight (kg) | | | | |
| < 2.5 | 6 (9) | 14 (11) | 0.85 (0.42–1.71) | – |
| ≥ 2.5 | 63 (91) | 116 (89) | 1 | |
| **Laboratory characteristics at admission** | | | | |
| C-reactive protein (CRP) | | | | |
| < 10 mg/l | 37 (39) | 71 (44) | 1 | |
| ≥ 10 mg/L | 58 (61) | 89 (56) | 1.15 (0.82–1.60) | – |
| White blood cell count | | | | |
| ≤ 12 x10^9/l | 42 (56) | 88 (64) | 1 | |
| > 12 x10^9/l | 33 (44) | 49 (36) | 1.24 (0.86–1.79) | – |
| **HIV *status*** | | | | |
| Exposed | 17 (17) | 32 (19) | 1.07 (0.69–1.66) | 0.92 (0.61–1.41) |
| Negative | 61 (63) | 128 (76) | 1 | 1 |
| Positive | 19 (20) | 9 (5) | **2.10 (1.51–2.92)**\*\*\* | **1.59 (1.08–2.33)**\*\* |
| Tested positive for malaria | | | | |
| No | 45 (85) | 42 (82) | 1 | |
| Yes | 8 (5) | 9 (18) | 0.90 (0.52–1.57) | – |
| **Comorbidities** [d] | | | | |
| Tuberculosis | | | | |
| Not Presumptive | 77 (75) | 171 (92) | 1 | 1 |

(*Continued*)

**Table 4.** (Continued)

| Variable | Pneumonia status during hospitalization[a] Pneumonia (N = 103) | Pneumonia status during hospitalization[a] No Pneumonia (N = 186) | Crude PR | Adjusted PR |
|---|---|---|---|---|
| Variable | n (%) | n (%) | | |
| Presumptive | 26 (25) | 15 (8) | **2.04 (1.51–2.75)***** | **1.73 (1.23–2.43)**** |
| **Clinical characteristics during hospitalization** | | | | |
| Pulse | | | | |
| Normal | 91 (90) | 159 (92) | 1 | |
| Abnormal | 10 (10) | 13 (8) | 1.19 (0.72–1.95) | – |
| Use of Resomal | | | | |
| No | 19 (18) | 42 (23) | 1 | |
| Yes | 84 (82) | 144 (77) | 1.18 (0.78–1.78) | – |
| Use of Nasogastric tube | | | | |
| No | 5 (53) | 127 (70) | 1 | 1 |
| Yes | 48 (47) | 55 (30) | **1.54 (1.14–2.09)**** | **1.38 (1.00–1.89) **** |
| Tested positive for malaria | | | | |
| No | 45 (85) | 42 (82) | 1 | |
| Yes | 8 (5) | 9 (18) | 0.90 (0.52–1.57) | – |
| Duration of hospital stay | | | | |
| 0 to 14 days | 21 (20) | 83 (45) | 1 | 1 |
| 15 days or more | 82 (80) | 103 (55) | **2.19 (1.44–3.32)***** | **2.16 (1.39–3.35)**** |

[t]T-test statistic, *P<0.1, **p<0.05, ***p<0.001
Abbreviations: WHZ, weight for height z score, HAZ, height for age Z score,
MUAC, mid-upper arm circumference, IQR, interquartile range, BF, breastfeeding.
[a] proportions were compared using $\chi^2$ test and continuous variables were compared using [t]T-test statistic.
[b] Immunisation status up-to-date: have completed receiving the recommended PCV 3.
[c] Breastfeeding status: we considered children less than 2 years who expected to still be breastfeeding by WHO standards.
[d] Comorbidities: a disease or medical condition that is simultaneously present in severely malnourished children who are the participants.

the prevalence lower than the previous study. Therefore, to improve on clinical diagnosis of pneumonia among children with SAM, it is advisable to do a CXR for children presenting with respiratory symptoms.

Notably, the prevalence of pneumonia in the current study is still high and similar to evidence before and after COVID 19. COVID 19 caused economic disruptions which led to increased food insecurity in households, disrupted food supply and breast feeding practices contributing to increased levels of malnutrition [31] However, after COVID 19 the economic disruptions reduced and the situation normalized. The current study was done prior to COVID 19 among children with severe acute malnutrition which is an important risk factor for pneumonia among children under 5 years [5]. And yet, the burden of malnutrition in our setting is still high (3.2%) according to the recent UDHS 2022 [4]. Therefore, findings in this study reflect the current situation of pneumonia among children in this age group and conclusions have significance in daily clinical practice.

There observations that have reported a decline in pneumonia admissions following introduction of PCV vaccination in Africa [32,33] and Uganda [34]. However, pneumonia is still reported to be the leading cause of death in children under 5 years in this age group [35]. In Uganda, PCV was introduced during the data collection period of the current study. However, hospital based pneumonia studies conducted among severely malnourished children

after introduction of PCV vaccine in Uganda have reported a prevalence of 26.8% (Mbarara) and 37.6% (Jinja) [9,10] which is similar with prevalence of 28% in the current study. In addition, the decline reported in pneumonia admissions may not be the same in malnourished children due to presence of other pneumonia-causative organisms that PCV does not protect against such as klebsiella and viral infections like CMV. Furthermore, the PCV vaccination (PCV3) vaccination coverage in Uganda is 72% leaving many children under 5 at risk of pneumonia [4].

**Factors associated with prevalence of pneumonia.** We found that female sex, HAZ scores of < -3SD (stunting), and presumptive TB were factors associated with pneumonia at admission in our analysis.

In the current study, the prevalence of pneumonia was higher among female children compared to males. Our findings are different from other studies which have found that male children are more susceptible to getting infections like pneumonia compared to females [36]. The possible reasons behind the high susceptibility of pneumonia in females than in males could arise from severe morbidity and socioeconomic factors. The female children may come with severe forms of SAM hence severe immunosuppression making them more prone to infections than males. Severe forms of SAM among females could be due to gender bias in food and health care allocation among children by the parents [37]. Parents particularly mothers may give higher attention to male children by giving them greater quantities of higher quality nutritious food and promptly taking them to a health facility when they are sick compared to female children [37,38]. Therefore, we recommend further studies to examine the sex differences in prevalence of pneumonia to broaden our understanding of correlates of pneumonia among children with SAM.

Notably, HAZ scores of < -3SD (stunting) was an important factor associated with both prevalence and incident pneumonia in the current study. From our study, children who had stunting, had a higher prevalence and incident pneumonia during hospitalization compared to those without. Our observations are similar to findings from other studies where stunting is associated with pneumonia [36,38]. In the study by Kumdin et al the odds of stunting were 3.6 times higher among children with pneumonia [38]. Stunting as a severe form of malnutrition has long term sequels on lung growth and development predisposing the child to getting pneumonia [38]. In addition, stunting is reported to increase risk of treatment failure and prolong course of recovery in children with pneumonia [39]. Therefore, to broaden our understanding of how stunting relates with pneumonia, we recommend further studies in these area.

In the current study, prevalence and incident pneumonia was higher among children with features of presumptive TB. The findings in the current study are similar to other studies where pneumonia is reported to be a strong correlate of TB in children with SAM [40,41]. In a study by Atalell et al, the odds of being infected with TB were 2.8 times higher among children with SAM and pneumonia as a comorbidity [40]. Children with SAM are prone to infections like *Mycobacteria Tuberculosis* causing pneumonia because of the impaired cell mediated immunity which is the principal host defense against TB.

Notably, some children with presumptive TB in the current study did not present with pneumonia at admission however they developed incident pneumonia during hospitalization. The possible explanation is some patients with presumptive TB could have responded to the initial antibiotics given due to bacterial co-infection while others failed to mount an immune response against TB due to severe immune suppression. However, as the immune system recovered during hospitalization they mounted an immune response causing a clinical manifestation of TB presenting as pneumonia. Therefore, malnourished children with

presumptive TB could be monitored and evaluated for incident pneumonia as a potential comorbidity during hospitalization.

Other factors associated with pneumonia in previous studies including breastfeeding were not reported in this study. This may be attributed to how data on these variable was collected so it could not be correctly interpreted contrary to what we know.

### Incident pneumonia and their associated factors

**Incident pneumonia.** Our results show incident pneumonia among children hospitalized with SAM was 356 per 1000 hospital admissions of children with SAM. To our knowledge, there is hardly any documentation on incident pneumonia among children hospitalized with SAM. However, previous studies have documented incident rate of pneumonia among general pediatric populations to range from 7% to 84% [22,42,43]. The differences in patient population, study design and case definition of incident pneumonia may explain this variability with the current study. Previous studies were conducted among general pediatric populations of children less than 5 years which includes both the well and poorly nourished children [22,42]. And yet our study focuses on children with SAM which is an important risk factor for incident pneumonia during hospitalization. Furthermore, incident pneumonia in the current study varied by study designs with the previous studies. In one previous study, incident pneumonia was determined by routine point prevalence survey for infection prevention in hospital settings [22] which underestimates incident pneumonia. And yet, the current study utilized data from the ProbiSAM trial database which was collected prospectively with children being monitored daily to identify incident cases of pneumonia. Other previous studies were retrospective case control studies where charts were reviewed causing over estimation of incident pneumonia due to recall bias [42].

The current findings of incident pneumonia among children with SAM is high. And yet, these children are supposed to be on their way to recovery from infections like pneumonia since they are receiving antibiotic and dietary treatment. The high incidence of pneumonia could be explained by some reasons. The incident pneumonia may be high considering that the study was conducted among malnourished children which is a risk factor of pneumonia among children less than 5 years [12,15]. In addition, the causative organisms identified for incident pneumonia are reported to be resistant to the common antibiotics used in our set up [15]. Furthermore, therapeutic feeds given to malnourished children stimulate recovery of the immune system to mount an appropriate immune response against organisms that are initially silent due to severe immunosuppression to manifest as pneumonia. Notably, the risk of death among children with incident pneumonia is reported to be high compared to those who come in with pneumonia at admission [15]. Therefore, we recommend further studies to explore the possible causative organisms and outcomes including mortality for incident pneumonia. This will enable giving appropriate treatment and reduce mortality related to pneumonia among children with SAM. Furthermore, national guidelines could be revised to include criteria to monitor, detect and treat children with incident pneumonia to substantially reduce mortality among children with SAM.

**Factors associated with incident pneumonia during hospitalization.** In the current study, age, stunting, HIV positive status, presumptive TB, use of a nasogastric tube for feeding and prolonged hospitalization were risk factors for developing incident pneumonia during hospitalization.

The findings in the current study show that young children aged between 6 to 12 months, were at a higher risk of incident pneumonia during hospitalization. And yet, there is limited data on how age relates with incident pneumonia in children with SAM. However,

previous studies on prevalence of pneumonia in children have reported younger age as a risk factor for pneumonia [13,44,45]. The odds of developing pneumonia among younger children less than 12 months was reported to be 2.5 and 3.0 times higher in Ethiopian and Ugandan studies respectively [13,44]. These findings are similar to the current study where young children aged between 6 to 12 months, were at a higher risk of incident pneumonia during hospitalization. The young infants are prone to pneumonia because of their weak immune system that allows progression of the upper respiratory infection to the lungs causing pneumonia.

In the current study, HIV positive children, had a higher risk of incident pneumonia during hospitalization. Our findings are similar to previous studies that have reported HIV infection as a risk factor for pneumonia among malnourished children [46,47]. Both HIV and malnutrition elicit dysfunction in the immune system and promote increased vulnerability of the host to infections like incident pneumonia. Previous studies have also indicated that HIV positive children with pneumonia have poor outcomes including mortality and slow nutritional recovery [46]. Therefore, in the context of high burden of HIV and malnutrition in our set up, incident pneumonia should be recognized as priority for detailed investigation and treatment given the heightened risk of death associated with it.

From the current study, children who used a nasogastric tube (NGT) for feeding during hospitalization were at risk of incident pneumonia. Our findings are similar with a previous study where NGT feeding was associated with aspiration of gastric contents leading to a high incidence of Gram-negative pneumonia [48]. On the contrary, gastric tube feeding is supposed to protect against aspiration. However, aspiration may occur as a complication and it is associated with adverse outcomes like mortality [49]. The possible mechanism for aspiration in malnourished children bearing an NGT is muscle weakness and relaxation of oesophageal sphincters causing aspiration. This may be coupled with a standard practice in malnourished children of frequent feeding day and night which is sometimes forceful because the children may have a poor appetite.

In the current study, prolonged hospitalization was a risk factor for incident pneumonia during hospitalization. This finding is similar with previous studies that have reported incident pneumonia to be strongly associated with prolonged hospital stay [12,15]. During prolonged hospital stay, the normal barriers against infection are lost resulting in increased colonization of the respiratory tract by potential pathogens that cause incident pneumonia [12,15]. In addition, prolonged hospital stay exposes the child to nosocomial infection in overcrowded settings like the case with our inpatient care wards. The organisms identified to commonly cause incident pneumonia are usually resistant to the common antibiotics used in our set up leading to fatal outcomes [15]. These findings highlight the need for clinicians to closely monitor children hospitalized for SAM for incident pneumonia so that prompt treatment and prevention of infection is done to reduce mortality. Furthermore, the policy makers may need to revise and include guidelines for the management of these children.

## Study strengths and limitations

The strengths of our study include: the large sample size that was sufficient to answer the study questions. The prospective study design that enabled us to establish temporal relationship between incident pneumonia and associated factors during hospitalization. The study investigated factors associated with both prevalence and incident pneumonia among children hospitalized with SAM. These factors may be considered when designing/reviewing management protocols for pneumonia among children with SAM. The secondary data set

used in the analysis was part of a randomized controlled trial where there was frequent monitoring of participants by well qualified clinicians the paediatricians who were part of the study.

However, our study was not devoid of limitations usually associated with retrospective analysis. Random error is a common challenge associated with a secondary dataset. Nonetheless, random error associated with secondary data set was minimized by the large sample size and population size sampling procedure used in randomized controlled trials. Selection bias is another inherent limitation of existing datasets. To minimize selection bias, we did sensitivity analysis for variables that had missing data. We compared models with or without missing data and since we found no differences in the findings, we reported findings with complete data. In the current study we used clinical criteria versus radiological criteria for the diagnosis of pneumonia. However, we do not think this could have impacted our study because the current WHO guidance is to routinely use clinical signs to make a diagnosis of pneumonia. In addition, children with SAM, may not have radiological signs of pneumonia yet they actually have pneumonia. Furthermore, the dataset analyzed for this study was generated for a different purpose rather than this specific study. However, from the review of the main ProbiSAM trial protocol pneumonia was one of the secondary objectives so there was a deliberate effort to identify signs and symptoms of pneumonia. The study setting was a tertiary hospital, therefore, our findings may not be generalizable to a different setting.

## Conclusion

This study identified that prevalence of pneumonia among children hospitalized with SAM is high and similar to previous hospital based studies conducted done in different settings in Uganda. Furthermore, the current study identified that malnourished children are at high risk of incident pneumonia during hospitalization. This implies the burden of pneumonia among children hospitalized with SAM is still significant despite efforts done to prevent, protect and treat pneumonia among children less than 5 years.

Stunting and presumptive Tuberculosis were identified as independent significant factors associated with both prevalence and incident pneumonia among children hospitalized with severe acute malnutrition in the current study.Other additional factors identified as independent significant factors associated with pneumonia among children hospitalized with severe acute malnutrition in the current study include; 1) female sex for prevalence of pneumonia 2) Younger age, HIV infection, use of nasogastric tube for feeding and prolonged hospitalization for incident pneumonia. Early recognition and management of these factors could improve outcomes of under-five children hospitalized with pneumonia and severe acute malnutrition, particularly in resource-limited areas.

Considering the risk of incident pneumonia among children hospitalized with SAM, there is urgent need to review guidelines to monitor, detect and treat incident pneumonia to substantially reduce mortality associated with it. Furthermore, we highly recommend studies to explore the possible causative organisms and outcomes including mortality for incident pneumonia among children with SAM. Studies to explore the relationship between stunting and pneumonia are also required.

## Acknowledgments

We would like to thank study participants, the entire study team for the ProbiSAM trial for their unreserved contribution during data collection. We gratefully acknowledge the principal investigator for ProbiSAM trial Professor Henrik Friis who allowed us to utilize

the data for analysis and write the manuscript. We acknowledge specifically Dr. Benedikte Grenov who was part of the ProbiSAM trial study team for the excellent support in the process of creating, organizing and maintaining data sets so they can be accessed and used for analysis.

## Author contributions

**Conceptualization:** Harriet Nambuya, Ezekiel Mupere, Nicolette Nabukeera-Barungi, Jolly General Kamugisha, Maren JH Rytter, Rebecca Nantanda.

**Data curation:** Harriet Nambuya, Ezekiel Mupere, Nicolette Nabukeera-Barungi, Paul Mubiri, Rebecca Nantanda.

**Formal analysis:** Harriet Nambuya, Ezekiel Mupere, Nicolette Nabukeera-Barungi, Paul Mubiri, Jolly General Kamugisha, Maren JH Rytter, Rebecca Nantanda.

**Investigation:** Harriet Nambuya, Ezekiel Mupere, Nicolette Nabukeera-Barungi, Paul Mubiri, Jolly General Kamugisha, Maren JH Rytter, Rebecca Nantanda.

**Methodology:** Harriet Nambuya, Ezekiel Mupere, Nicolette Nabukeera-Barungi, Paul Mubiri, Jolly General Kamugisha, Maren JH Rytter, Rebecca Nantanda.

**Project administration:** Harriet Nambuya, Nicolette Nabukeera-Barungi, Maren JH Rytter.

**Supervision:** Ezekiel Mupere, Nicolette Nabukeera-Barungi, Rebecca Nantanda.

**Writing–original draft:** Harriet Nambuya, Nicolette Nabukeera-Barungi.

**Writing–review & editing:** Harriet Nambuya, Ezekiel Mupere, Nicolette Nabukeera-Barungi, Paul Mubiri, Rebecca Nantanda.

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
