## [Decision Letter · Decision Letter 0]

24 Apr 2024

PONE-D-23-36220Prevalence, incidence and associated factors of pneumonia among severely malnourished children hospitalized in a tertiary hospital in Uganda: Mixed study designPLOS ONE

Dear Dr. Nambuya,

Thank you for submitting your manuscript to PLOS ONE. After careful consideration, we feel that it has merit but does not fully meet PLOS ONE’s publication criteria as it currently stands. Therefore, we invite you to submit a revised version of the manuscript that addresses the points raised during the review process.

**Please review the article carefully and incorporate their suggestions. please also adress the timeliness of the study **

We look forward to receiving your revised manuscript.

Kind regards,

George Kuryan

Academic Editor

PLOS ONE

Note from Kalarani A (plosone@plos.org):This manuscript reports a study of data collected during a clinical trial. The main trial is registered at ProbiSAM(ISRCTN16454889) . If a statistical review is needed, email plosone@plos.org and PLOS staff will assign one of our statistical advisors (http://journals.plos.org/plosone/s/advisory-groups#loc-statistical-advisors).

Reviewers' comments:

Reviewer's Responses to Questions

**Comments to the Author**

1. Is the manuscript technically sound, and do the data support the conclusions?

Reviewer #1: Yes

Reviewer #2: Partly

2. Has the statistical analysis been performed appropriately and rigorously? 

Reviewer #1: Yes

Reviewer #2: Yes

3. Have the authors made all data underlying the findings in their manuscript fully available?

Reviewer #1: No

Reviewer #2: No

4. Is the manuscript presented in an intelligible fashion and written in standard English?

Reviewer #1: No

Reviewer #2: No

5. Review Comments to the Author

Reviewer #1: I have read your manuscript with keen interest. The document is well-written and addresses an important tropical paediatric problem—the burden of pneumonia and associated factors in children admitted with severe acute malnutrition. However, I have a few minor comments:

1. There are several typographical errors throughout the document. For example, line 52, consider saying risk of pneumonia, rather than risk of incidence of pneumonia. Line 68, consider ‘pneumonia was associated with 740,180 childhood deaths’, rather than ‘pneumonia killed 740,180 children’. Please correct these.

2. Line 443, consider that ‘M.tuberculosis rather than tuberculosis is one of the causative organisms..’. Please correct.

3. Within Discussion, consider incident pneumonia, rather than incidence of pneumonia. Please correct.

4. Sample size estimation: the statement on what was the sample computed to detect is incomplete. Please clarify.

5. Limitations: consider that the dataset analysed for this study was generated for a different purpose rather than this specific study. Secondly, consider that the study setting is a tertiary hospital, meaning that the findings may not be generalizable to a different setting.

Reviewer #2: Thank you for choosing very interesting topics. Though there are lots of publication related to this topic.

The manuscript is well organized and analysis was done very meticulously. Several observations are there for this manuscript,

1. We know that young age, any kind of malnutrition, fever etc are risk factors for developing the pneumonia. So, the findings of your manuscript are not exclusive in this field.

2. Important observation is data collection period. It was 10th March 2014 to 8th July 2015. Old data set which might not reflecting the current situation. By this period COVID has changed the scenario. It has changed the economic condition, immunity, disease pattern etc.

3. in Uganda, PCV was launched in 2014. So, situation might be changed in terms of prevalence and incidence by this period.

4. It would be more catchy if we have the mortality data for community acquired and hospital acquired pneumonia cases.

5. Result: How can you use “Tested positive for malaria” under clinical characteristic during hospitalization in Table 3 and 4? It is part of investigation. Fever is one of the presentations of pneumonia, how could it be a risk factor for pneumonia?

Variables selection for identifying the risk factors was not proper.

6. Result: In table 3 we have observed that total participants were 289, edema was present in 200 cases. So, we are getting improper WHZ for edematous cases, as weight is more in edematous cases.

7. Discussion: The discussion is too long. For both prevalence and incidence almost, similar observations were described twice. So, redundancy could be avoided with proper use of language.

6. PLOS authors have the option to publish the peer review history of their article (what does this mean?). If published, this will include your full peer review and any attached files.

Reviewer #1: **Yes: **Dr Charles Obonyo

Reviewer #2: No

---

## [Author Response · Author response to Decision Letter 1]

20 Jul 2024

Dear George Kuryan

Academic Editor

PLOS ONE

We would like to thank the editorial team and the reviewers for reviewing our submission of the above referenced manuscript to PLOS ONE journal for consideration. We have given careful consideration to all the comments /questions from the reviewers.

Have uploaded attachment's of

1.Response to reviewers

2.Revised Manuscript with track changes

3.Clean copy of Manuscript

as requested

Looking foward to positive response

---

## [Decision Letter · Decision Letter 1]

26 Jan 2025

PONE-D-23-36220R1Prevalence, incidence and associated factors of pneumonia among severely malnourished children hospitalized in a tertiary hospital in Uganda: Mixed design StudyPLOS ONE

Dear Dr. Nambuya,

Thank you for submitting your manuscript to PLOS ONE. After careful consideration, we feel that it has merit but does not fully meet PLOS ONE’s publication criteria as it currently stands. Therefore, we invite you to submit a revised version of the manuscript that addresses the points raised during the review process.

Please submit your revised manuscript Mar 12 2025 11:59PM. If you will need significantly more time to complete your revisions, please reply to this message or contact the journal office at plosone@plos.org. Please include the following items when submitting your revised manuscript:

We look forward to receiving your revised manuscript.

Kind regards,

Frederick Quinn

Academic Editor

PLOS ONE

Journal Requirements:

Reviewers' comments:

Reviewer's Responses to Questions

**Comments to the Author**

1. If the authors have adequately addressed your comments raised in a previous round of review and you feel that this manuscript is now acceptable for publication, you may indicate that here to bypass the “Comments to the Author” section, enter your conflict of interest statement in the “Confidential to Editor” section, and submit your "Accept" recommendation.

Reviewer #2: All comments have been addressed

Reviewer #3: All comments have been addressed

2. Is the manuscript technically sound, and do the data support the conclusions?

Reviewer #2: Partly

Reviewer #3: Yes

3. Has the statistical analysis been performed appropriately and rigorously? 

Reviewer #2: Yes

Reviewer #3: Yes

4. Have the authors made all data underlying the findings in their manuscript fully available?

Reviewer #2: No

Reviewer #3: Yes

5. Is the manuscript presented in an intelligible fashion and written in standard English?

Reviewer #2: Yes

Reviewer #3: Yes

6. Review Comments to the Author

Reviewer #2: It is a well written manuscript on an important health problem. Few observations are there for the manuscript.

Introduction:

Authors used old statistics related pneumonia and malnutrition. Recent statistics should be used.

Line 78-79 and line 88-89 are providing almost similar information. Redundancy should be avoided.

Method: It is well organized.

Result:

Table 1:organize the values for variable Septicemia.

Discussion:

Line 430-432: Therefore, CXR should be done for all……….these children. This advice should not be applied directly rather author can use judicious selection of candidates for xray.

Line 433: Please correct the l ine.

Still discussion is long. And also, there is redundant information.

Reviewer #3: It is a relevant study. The reviewer comments have been addressed. However, the prevalence of any disease with data which is 10 years old is questionable. That aspect of the study may not be relevant and in fact may be misleading - especially since the pandemic and addition of vaccines in the national schedule. Otherwise, this is a well written study.

I would suggest removing the prevalence from the study title and remove the focus from the prevalence.

7. PLOS authors have the option to publish the peer review history of their article (what does this mean?). If published, this will include your full peer review and any attached files.

Reviewer #2: No

Reviewer #3: No

---

## [Author Response · Author response to Decision Letter 2]

18 Mar 2025

Dear Editor,

thank you so much for considering our manuscript for review by PLOS ONE.We have attended all the comments and questions raised carefully. I have attached the documents below for details

1.rebuttal letter (response to reviewers)

2.revised manuscript with track changes

3.Manuscript (revised manuscript clean copy)

Thank you so much for your consideration

kind regards

Harriet Nambuya (lead author)

---

## [Decision Letter · Decision Letter 2]

20 May 2025

PONE-D-23-36220R2Prevalence, incidence and associated factors of pneumonia among severely malnourished children hospitalized in a tertiary hospital in Uganda: Mixed design StudyPLOS ONE

Dear Dr. Nambuya,

Thank you for submitting your manuscript to PLOS ONE. After careful consideration, we feel that it has merit but does not fully meet PLOS ONE’s publication criteria as it currently stands. Therefore, we invite you to submit a revised version of the manuscript that addresses the points raised during the review process.

Please submit your revised manuscript by Jul 04 2025 11:59PM. If you will need significantly more time to complete your revisions, please reply to this message or contact the journal office at plosone@plos.org. Please include the following items when submitting your revised manuscript:

We look forward to receiving your revised manuscript.

Kind regards,

Frederick Quinn

Academic Editor

PLOS ONE

Journal Requirements:

Reviewers' comments:

Reviewer's Responses to Questions

**Comments to the Author**

1. If the authors have adequately addressed your comments raised in a previous round of review and you feel that this manuscript is now acceptable for publication, you may indicate that here to bypass the “Comments to the Author” section, enter your conflict of interest statement in the “Confidential to Editor” section, and submit your "Accept" recommendation.

Reviewer #4: All comments have been addressed

Reviewer #5: (No Response)

2. Is the manuscript technically sound, and do the data support the conclusions?

Reviewer #4: Yes

Reviewer #5: Yes

3. Has the statistical analysis been performed appropriately and rigorously? 

Reviewer #4: Yes

Reviewer #5: Yes

4. Have the authors made all data underlying the findings in their manuscript fully available?

Reviewer #4: Yes

Reviewer #5: Yes

5. Is the manuscript presented in an intelligible fashion and written in standard English?

Reviewer #4: Yes

Reviewer #5: Yes

6. Review Comments to the Author

Reviewer #4: (No Response)

Reviewer #5: Title: The title is clear and conveys the essence of the authors’ work.

Although the title states, “….Prevalence, incidence and associated factors of pneumonia among severely malnourished children hospitalized in a tertiary hospital in Uganda: Mixed study design…”, It implies that the authors set out to study Severe malnourishment. However their work specifically describes severe acute malnutrition (SAM). Are these the same? I suggest they stick to the same nomenclature in the title..

Background: The background and literature is exhaustive and introduces the main

research idea adequately.

Methods: You state in the title that the study was conducted at a tertiary hospital in Uganda yet you specify here that The ProbiSAM trial was conducted at Mwanamugimu Nutritional Unit Mulago National Referral and Teaching Hospital between 10th March 2014 to 8th July 2015. You also confirm that it serves referral cases from all over the country. Why don’t you refer to this in your title?

Whereas the authors state that the average monthly attendance for inpatients was 120 children aged 1 months to 12 years, 75% of whom are aged 6 to 59 months as the basis for their inclusion of data to be analyzed. This reason is weak and a more detailed explanation and explanation for its potential bias due to the arbitrary cutoff is recommended.

In discussing procedures at Admission, the authors do not describe what was done in this work relative to the parent study. Please confirm and correctly reference if the procedures described have been reported elsewhere especially by the ProbiSAM trial.

In the title, a Mixed methods design was advance. I expected both qualitative and quantitative study method, I only see the later. Kindly confirm why there is need to qualify your work as ‘mixed methods design’- preferably with a reference.

Ethical Approval: It is stated that Human Subject’s Approval; was obtained…..Can you confirm if this is intended to mean Ethical Approval?

Please confirm and state if the Parent study received Institutional Review Board approval in Uganda.

It is stated that, “Written informed consent was obtained from all care givers before enrolment into the study”. Was this for the Parent study? If so, clearly state it. Also confirm if the minors provided assent and whether consent for use of secondary data was sought and granted during the parent study. Otherwise, a waiver of consent from the REC should have been obtained.

Results: In their results the authors do not discuss the role of probiotics which were administered in the parent study to their outcome. This would be a key variable to consider as part of the analysis plan.

Line 281, States that, “Most of the children were aged below 24 months” Can you qualify this with a statistic if it’s important. This generally applies to the entire results section – review the use of generalized and unsupported narratives in the results section.

Were there any microbiological tests done to confirm causative agents for pneumonia in addition to MTB?

Whereas you discuss mortality as part of your work (line 479), you do not show any mortality outcomes/data as part of the results. Was this data collected?

Discussion: The authors’ claim for this to be the first study to report incidence in such a large enough sample size is commendable and a strength for their work.

Please emphasize the discussion regarding the time the study was conducted in relation to efforts being made to prevent, protect and treat pneumonia to reduce the morbidity and mortality related with it among children less than 5 years by the year 2025. Are your results enough to justify/support your discussion since the work was done between March 2014 and July 2015?

Line 443: …is M. tuberculosis (the causative organism of TB…), Please note that M. tuberculosis should be always written underlined or italicized.

Consider, line 444: “Prone to infections like M. tuberculosis because…of the…”

Line 446: I suggest you state, “ Could be” and not “Should” since this work is only part of the body of evidence. Consider this for the entire discussion section.

Line 479: Your discussion of mortality is not supported by your data and results. Please review.

7. PLOS authors have the option to publish the peer review history of their article (what does this mean?). If published, this will include your full peer review and any attached files.

Reviewer #4: **Yes: **Dr Kankya Clovice

Reviewer #5: No

---

## [Author Response · Author response to Decision Letter 3]

30 Jul 2025

We would like to thank the editorial team and the reviewers for reviewing our submission of the above referenced manuscript to PLOS ONE journal for consideration. We have given careful consideration to all the comments /questions from the reviewers.

I have attached details of the Rebuttal letter, Manuscript clean copy and Revised Manuscript with track changes awaiting feedback after Resubmission. In addition we have attached approval letter for the probiSAM trial,letter from the Principal investigator allowing us to use the secondary data from ProbiSAM and approval of the current protocol.

---

## [Decision Letter · Decision Letter 3]

20 Aug 2025

Prevalence, incidence and associated factors of pneumonia among severely malnourished children hospitalized in Mulago National Referral hospital, Uganda

PONE-D-23-36220R3

Dear Dr. Nambuya,

We’re pleased to inform you that your manuscript has been judged scientifically suitable for publication and will be formally accepted for publication once it meets all outstanding technical requirements.

Kind regards,

Frederick Quinn

Academic Editor

PLOS ONE

Additional Editor Comments (optional):

Reviewers' comments:

Reviewer's Responses to Questions

**Comments to the Author**

1. If the authors have adequately addressed your comments raised in a previous round of review and you feel that this manuscript is now acceptable for publication, you may indicate that here to bypass the “Comments to the Author” section, enter your conflict of interest statement in the “Confidential to Editor” section, and submit your "Accept" recommendation.

Reviewer #5: All comments have been addressed

2. Is the manuscript technically sound, and do the data support the conclusions?

Reviewer #5: Yes

3. Has the statistical analysis been performed appropriately and rigorously? 

Reviewer #5: Yes

4. Have the authors made all data underlying the findings in their manuscript fully available?

Reviewer #5: Yes

5. Is the manuscript presented in an intelligible fashion and written in standard English?

Reviewer #5: Yes

6. Review Comments to the Author

Reviewer #5: Version 3 captures the recommendations to my satisfaction as stated in the rebuttal letter. Authors have changed the title to reflect the study area and responded to all other recommendations accordingly.

7. PLOS authors have the option to publish the peer review history of their article (what does this mean?). If published, this will include your full peer review and any attached files.

Reviewer #5: No

---

## [Editor Report · Acceptance letter]

PONE-D-23-36220R3

PLOS ONE

Dear Dr. Nambuya,

I'm pleased to inform you that your manuscript has been deemed suitable for publication in PLOS ONE. Congratulations! Your manuscript is now being handed over to our production team.

Kind regards,

on behalf of

Dr. Frederick Quinn

Academic Editor

PLOS ONE